# Allergenic Shrimp Tropomyosin Distinguishes from a Non-Allergenic Chicken Homolog by Pronounced Intestinal Barrier Disruption and Downstream Th2 Responses in Epithelial and Dendritic Cell (Co)Culture

**DOI:** 10.3390/nu16081192

**Published:** 2024-04-17

**Authors:** Marit Zuurveld, Anna M. Ogrodowczyk, Sara Benedé, Rebecca Czolk, Simona Lucia Bavaro, Stefanie Randow, Lidia H. Markiewicz, Barbara Wróblewska, Elena Molina, Annette Kuehn, Thomas Holzhauser, Linette E. M. Willemsen

**Affiliations:** 1Division of Pharmacology, Utrecht Institute for Pharmaceutical Science, Faculty of Science, Utrecht University, 3584 CG Utrecht, The Netherlands; m.zuurveld@uu.nl; 2Department of Immunology and Food Microbiology, Institute of Animal Reproduction and Food Research, Polish Academy of Sciences, 10-748 Olsztyn, Poland; 3Department of Bioactivity and Food Analysis, Instituto de Investigación en Ciencias de la Alimentación (CIAL, CSIC-UAM), 28049 Madrid, Spain; 4Department of Immunology, Ophthalmology and ENT, Faculty of Medicine, Complutense University of Madrid, 28040 Madrid, Spain; 5Department of Infection and Immunity, Luxembourg Institute of Health, 4354 Esch-sur-Alzette, Luxembourg; 6Faculty of Science, Technology and Medicine, University of Luxembourg, 1359 Kirchberg, Luxembourg; 7Institute of Sciences of Food Production, National Research Council (Ispa-Cnr), 70126 Bari, Italy; 8Division of Allergology, Paul-Ehrlich-Institut, 63225 Langen, Germany

**Keywords:** epithelial barrier, mucosal immunology, sensitizing allergenicity, food allergy, tropomyosins

## Abstract

Background: Tropomyosins (TM) from vertebrates are generally non-allergenic, while invertebrate homologs are potent pan-allergens. This study aims to compare the risk of sensitization between chicken TM and shrimp TM through affecting the intestinal epithelial barrier integrity and type 2 mucosal immune activation. Methods: Epithelial activation and/or barrier effects upon exposure to 2–50 μg/mL chicken TM, shrimp TM or ovalbumin (OVA) as a control allergen, were studied using Caco-2, HT-29MTX, or HT-29 intestinal epithelial cells. Monocyte-derived dendritic cells (moDC), cocultured with HT-29 cells or moDC alone, were exposed to 50 μg/mL chicken TM or shrimp TM. Primed moDC were cocultured with naïve Th cells. Intestinal barrier integrity (TEER), gene expression, cytokine secretion and immune cell phenotypes were determined in these human in vitro models. Results: Shrimp TM, but not chicken TM or OVA exposure, profoundly disrupted intestinal barrier integrity and increased alarmin genes expression in Caco-2 cells. Proinflammatory cytokine secretion in HT-29 cells was only enhanced upon shrimp TM or OVA, but not chicken TM, exposure. Shrimp TM enhanced the maturation of moDC and chemokine secretion in the presence or absence of HT-29 cells, while only in the absence of epithelial cells chicken TM activated moDC. Direct exposure of moDC to shrimp TM increased IL13 and TNFα secretion by Th cells cocultured with these primed moDC, while shrimp TM exposure via HT-29 cells cocultured with moDC sequentially increased IL13 expression and IL4 secretion in Th cells. Conclusions: Shrimp TM, but not chicken TM, disrupted the epithelial barrier while triggering type 2 mucosal immune activation, both of which are key events in allergic sensitization.

## 1. Introduction

Food allergic diseases are a growing health problem in Western societies. A recent systematic review and meta-analysis described that up to almost 10% of children are burdened with a physician-diagnosed food allergy [1]. Most food allergic reactions are induced by peanut-, cow’s milk-, egg-, soy-, wheat-, tree nut-, fish-, and shellfish-containing products [2], but numerous other food allergens have been identified [3]. Allergenic food proteins often possess intrinsic properties to promote allergic sensitization, including intestinal epithelial activation and permeabilization [4,5,6,7,8,9]. This mucosal immune activation can lead to the activation of dendritic cells (DC) and subsequent polarization of a type 2 T cell response which drives the development of allergen-specific IgE production and IgE-mediated clinical reactions upon subsequent encounter with the allergen [10]. Currently, there is a lack of in vitro models enabling discrimination between proteins in food products with a low or high risk for allergic sensitization [11,12]. 

Tropomyosins are functional proteins present in all eukaryotic cells and key regulators of muscle contraction [13]. Typically, in most vertebrate species, including chicken, these tropomyosins have a low sensitization and allergenicity risk [14,15]. However, tropomyosins that can be found in arthropods, such as crustaceans and insects, or in mollusks are important allergens. Tropomyosins contain an evolutionary highly conserved dimeric α-helical structure, shown by the 54–60% shared sequence between vertebrate and invertebrate tropomyosins [16]. Recently it was described that vertebrate (chicken) and invertebrate (shrimp) tropomyosin differ in gastric digestion, but both bound to IgE from shrimp-allergic patients. However, a positive skin prick test was only observed with shrimp tropomyosin in these patients [17]. In several types of existing assays, the allergic effector response is characterized and associated with allergenic potential that would indicate towards the risk of an allergic reaction. However, sensitizing allergenicity identifies the possible intrinsic property of a protein to activate type 2 driving immune cascades and, thus, increases the risk of allergic sensitization and food allergy development. The first step in this cascade is the breaching and/or activation of the epithelial barrier followed by activation of DC and a type 2 T cell response [18]. A previous study has shown this response for ovalbumin (OVA) as model allergen, which was able to activate intestinal epithelial cells (IEC) and/or monocyte-derived dendritic cells (moDC) [19]. 

This study aims to investigate the differential capacity of chicken TM (low sensitizing capacity) versus shrimp TM (high sensitizing capacity) in epithelial barrier disruption and activation, and in mucosal type 2 activation. Human in vitro models were used, including intestinal epithelial cells and blood-derived immune cells, which may provide tools to study the intrinsic sensitizing properties of current and future dietary proteins. 

## 2. Materials and Methods

### 2.1. Allergenic Proteins

High-allergenic recombinant black tiger shrimp tropomyosin (Pen m 1; shrimp TM) and low-allergenic recombinant chicken tropomyosin (chicken TM) were expressed in *E. coli* and purified as previously described [17,20]. The recombinant proteins were purified from LPS (as described in the Appendix A) and the final concentration of LPS was lower than 1 ng (10 EU)/mL for both TM proteins when used in a concentration of 50 μg/mL. The amino acid sequences of shrimp TM and chicken TM are presented in Appendix A. A detailed description of cloning, expression and isolation can be found in the Appendix A. A ‘control’ condition was used in every cell culture experiment; this control medium condition was not exposed to any additional protein component. OVA was used as an allergenic protein model known to induce activation of IEC and/or moDC to drive sequential mucosal type 2 responses [19,21]. 

### 2.2. Caco-2, HT-29MTX and HT-29 Cell Culture

Several human IEC model cell lines (colon carcinogenic origin) were used. Caco-2 cells were used for epithelial barrier studies; HT-29 cells were used to study epithelial cytokine responses. Caco-2 cells (passages 28–33 and 45–47) from the American Type Culture Collection (ATCC, USA) were cultured for 21 days in DMEM medium supplemented with 10% fetal bovine serum (FBS), 1% l-glutamine, 1% sodium pyruvate, 1% penicillin/streptomycin/amphotericin and 1% non-essential amino acids (all from Biowest, Nuaillé, France). The Caco-2 cells were seeded at a density of 1 × 10^5^ cells/well onto 12 well or 1.67 × 10^4^ cells/well onto 24 well transwell inserts (0.4 μm pore size; Costar, Washington, DC, USA; Corning, New York, NY, USA) and grown 2–3 weeks post confluency before use for barrier studies. HT29-MTX cells (methotrexate-selected HT29 clone, gifted by Dr. Giblin from Teagasc Food Research Centre, Moorepark, Fermoy, Ireland) were seeded in a 3:1 ratio (Caco-2:HT-29MTX) at a density of 3 × 10^4^ cells/well onto 12 well transwell inserts (0.4 μm pore size; Costar, Corning). 

HT-29 cells (passage 158–161), obtained from ATCC, were grown until ~80% confluency in 25 cm^2^ cell culture flasks was achieved. McCoy’s 5A medium (Gibco, Norristown, PA, USA), containing 10% heat-inactivated FBS, 1% penicillin and streptomycin (Sigma-Aldrich, London, UK) was used. Upon trypsinization, HT-29 cells were 5 times diluted based on surface area and seeded in transwell inserts (Corning Incorporated, New York, NY, USA) and cultured for 6 days into confluent layers to be used for coculture experiments.

Additionally, Caco-2 cells were seeded and cultured for 21 days, or HT-29 cells were seeded and cultured into confluent layers for 6 days in 96 wells plates. 

### 2.3. Transepithelial Electrical Resistance (TEER) and Cytotoxicity Assay

Barrier permeability of transwell cultured Caco-2 or mixed Caco-2/HT-29MTX monolayer, featuring characteristics of polarized enterocytes and mucus-producing cells, was measured using a Millicell ERS-2 electrical-resistance system (Millipore, Burlington, MA, USA) prior to and during the allergen exposure and TEER was expressed as ohm.cm^2^. 

The LDH (lactate dehydrogenase) activity cytotoxicity assay was carried out in the supernatants of exposed Caco-2 cells by using the CyQUANT™ LDH Cytotoxicity Assay Kit (Invitrogen, Waltham, MA, USA) according to the manufacturer’s instructions and expressed as an absorbance value (OD490-655 nm).

### 2.4. Caco-2 and PBMC Coculture

Healthy volunteer donor peripheral blood mononuclear cells (PBMCs) were isolated by density-gradient separation (Ficoll-Histopaque, GE Healthcare, Barcelona Spain) from heparinized venous blood samples from 5 non-allergic subjects obtained from the Institute of Food Science Research (CIAL, Spain) blood library. Written informed consent was obtained from all participants, according to the procedures approved by the Bioethics Committee of the CIAL. 

PBMCs (2 × 10^6^ cells/well) were added to the basolateral compartment of 12-well transwell plates where Caco-2 cells (at passage 28–33) where previously fully differentiated. Chicken TM, shrimp TM or OVA were apically added to the Caco-2 and PBMCs coculture and incubated for 8 h at 37 °C and 5% CO_2_. Caco-2 cells were preserved at −80 °C in RA-1 buffer (Macherey-Nagel, Düren, Germany) for gene expression analyses by qPCR (see Appendix A).

### 2.5. HT-29-moDC or moDC and Sequential DC/T Cell Coculture

This coculture model to study mucosal immunity was previously described in more detail [19]. In brief, healthy donor PBMC from volunteers, who had given written informed consent for research purposes, were obtained from the Dutch Blood bank (Amsterdam, The Netherlands). Monocytes and naïve Th cells were isolated using negative selection by magnetic beads. Monocytes were differentiated into immature moDC for 6 days using GM-CSF and interleukin (IL)4, and Th cells were stored in liquid nitrogen. After HT-29 cells reached confluence in the transwells, 5 × 10^5^ moDC were added to the basolateral compartment for 48h. The epithelial cells in the HT-29/moDC coculture were apically exposed to chicken TM or shrimp TM. Simultaneously, wells containing moDC without IEC, were also exposed to chicken TM or shrimp TM for 48h, allowing direct interaction with the moDC. Subsequently, moDC were collected for analysis by flow cytometry (FACS CantoII, BD Biosciences, Franklin Lakes, NJ, USA) and coculture with allogenic naïve T cells (stimulated with αCD3 and IL2). moDC and T cells (in a 1:10 ratio) were cocultured for 96 h. Cells were collected for flow cytometric analysis and supernatants were collected to measure cytokines using ELISA (R&D systems, Minneapolis, MN, USA) or multiplex array (Meso Scale Discovery, Rockville, MD, USA). 

### 2.6. Cytokine Measurements

Cytokine levels were determined in collected cell-free supernatants. Concentrations of IL33, TSLP, IL25, CCL20, CCL22, IL8 and IL13, IL4, IFNγ, IL10, IL17, IL21 and TNFα, were measured by commercially available ELISA kits (R&D systems, USA or Invitrogen, USA) according to the manufacturer’s instructions. To compare cytokine secretion after OVA, shrimp TM or chicken TM exposure of Caco-2 cells and HT-29 cells (cultured in 96 wells), a multiplex array was performed (Meso Scale Discovery, USA) to measure secreted levels of epithelial-derived IL33, TSLP, IL25, IL1α, IL1β, IL6, IL8 CCL20, CCL22 and TNFα in undiluted supernatant, according to the manufacturer’s instructions. 

### 2.7. Flow Cytometry

Flow cytometric analysis was performed on cocultured moDC and Th cells. After collection of the cells, the cells were washed with PBS, stained with Fixable Viability Dye eFluor 780 (eBioscience, San Diego, CA, USA) and nonspecific binding was blocked. Staining of moDC was performed using titrated amounts of CD11c-PerCP eFluor 710, HLA-DR-PE, CD80-FITC, CD86-PE-Cy7 and OX40L-APC. Staining of Th cells was performed with titrated volumes of CD4-PerCP-Cy5.5, CXCR3-AF488, CRTH2-APC and IL13-PE. All antibodies were purchased from eBioscience or BD Biosciences. Flow cytometric data was collected using a BD FACS CantoII (BD Biosciences, USA) and analyzed using FlowLogic software (Version 8, Inivai Technologies, Mentone, Australia).

### 2.8. Gene Transcription Analysis

Total RNA was isolated using an RNA Isolation Kit (Macherey-Nagel, Düren, Germany) and reverse transcribed into cDNA using PrimeScript RT Reagent Kit (Takara Bio Inc., Shiga, Japan). Primer pairs and thermal cycling conditions for qPCR assays are described in Table 1. Relative gene transcription was calculated by normalizing data to the transcription of the *Gadph* gene, using the 2^−ΔΔCT^ method.

### 2.9. Statistics

Statistical analysis was performed using Graphpad Prism (version 9.4.1). Data was analyzed using One-Way ANOVA, or Friedmann test when data did not fit a normal distribution. All conditions were compared to the control condition, therefore a Dunett’s post hoc test (if normally distributed) or Dunn’s post hoc test was performed. A Two-Way ANOVA was applied to analyze TEER data measured at several time points during protein exposure. *p* < 0.05 is considered statistically significant, and data is represented as mean ± standard error of mean (SEM) of n = 3 or n = 5 independent repeats of a full dataset per in vitro model. 

## 3. Results

### 3.1. Shrimp TM Decreases Epithelial Barrier and Induces Expression of Alarmins

First the effects of chicken TM, shrimp TM or OVA exposure on intestinal epithelial barrier integrity were investigated in transwell Caco-2 models. Forty-eight hour exposure of Caco-2 cells (Figure 1A) to chicken TM or OVA resulted in an overall increased TEER compared to medium controls for most concentrations (Figure 1B,D,E). By contrast, all shrimp TM doses, except for 2 μg/mL, resulted in an overall decrease in TEER (Figure 1C). However, at the 6 h timepoint only a decrease in TEER by 10, 25 and 50 μg/mL shrimp TM was observed, which was not detected for similar concentrations of chicken TM and OVA. Secretion of IL8, as marker of epithelial activation, was increased in all conditions (Figure 1F). Nonetheless, in this model some increase in cytotoxicity was also measured when exposing Caco-2 to all allergens except for the lowest concentrations used (Figure 1G). Subsequent experiments were performed with 50 μg/mL of tropomyosins to provoke an epithelial response, which allows comparison to the concentrations used for OVA-induced IEC activation. In addition, subsequent experiments did not reveal a decrease in viability upon exposure to 50 μg/mL tropomyosins (Appendix A).

Next, Caco-2/HT-29MTX cells were cultured in transwells and exposed to 50 μg/mL of chicken TM, shrimp TM or OVA for 8 h to validate the decrease in barrier integrity found in Caco-2 cells (Figure 1H). Similar to the Caco-2 cells, TEER was decreased after exposure to shrimp TM in Caco-2/HT-29MTX cultures compared to the medium control, and this was not observed for chicken TM or OVA (Figure 1I). In addition, a coculture was performed combining Caco-2 cells grown in 12 wells transwell plates with PBMCs in the basolateral compartment (Figure 1J). After 8 h of incubation with chicken TM, shrimp TM or OVA, expression of type-2-associated alarmins *Il33*, *Il25*, and *Tslp* was increased in Caco-2 cells after exposure to 25 μg/mL and 50 μg/mL shrimp TM (Appendix A; Figure 1K–M). By contrast, 50 μg/mL chicken TM exposure only enhanced *Il33* expression and OVA did not enhance mRNA expression of these alarmins (Figure 1K). In addition, PBMC-derived cytokines were measured but these remained below the detection limit. 

### 3.2. Cytokine and Chemokine Secretion Is Enhanced in HT-29 Cells after Allergen Exposure

In order to study chicken TM, shrimp TM or OVA-induced IEC activation and cytokine secretion, the effects on Caco-2 and HT-29 cell lines were compared using multiplex analyses. Cells were grown in 96 well plates and exposed for 48 h to 50 μg/mL of each protein (Figure 2A). No cytotoxicity was observed in these cultures (Appendix A). Despite increased mRNA expression of *Il33*, *Il25* and *Tslp* in transwell-grown Caco-2 cells after shrimp TM exposure (Figure 1K–M), the secretion levels of these cytokines were not elevated after 48 h exposure in the 96 wells plates (Figure 2B–D). Yet, CCL20 secretion was significantly increased in this condition (Figure 2G). On the other hand, enhanced secretion of IL25, IL1α, IL1β, CCL22, IL8, and TNFα was shown by HT-29 cells when exposed to shrimp TM, but not with chicken TM, which even lowered IL8 secretion (Figure 2C,E,F,H,I,K). Increased cytokine secretion by HT-29 cells was observed to a lesser extent after exposure to 25 μg/mL shrimp TM (Appendix A). OVA exposure also significantly enhanced secretion of ILβ, CCL22, and IL8 (Figure 2F,H,I). A dose–response experiment was performed in HT-29 cells and epithelial-derived mediators were measured by ELISA. Secretion of IL8 and CCL20 from HT-29 cells in response to either shrimp TM or OVA exposure dose-dependently increased, while chicken TM did not affect these mediators (Appendix A). 

### 3.3. Shrimp TM Induces moDC Activation in Presence and Absence of IEC

To study the sensitizing capacity of chicken TM and shrimp TM, these proteins were added to HT-29 cell/moDC (IEC-DC) coculture or moDC (DC) in absence of HT-29 cells for 48 h (Figure 3A). After exposure, chicken TM increased the proportion of moDC expressing the costimulatory receptor CD80 (Figure 3B) as well as CCL22 secretion (Figure 3E). A small increase in CCL20 secretion was observed (Figure 3F), but only during direct exposure of moDC. Exposure to shrimp TM also increased the percentage of moDC expressing CD80 (Figure 3B) and secretion of CCL20, CCL22, and IL8 (Figure 3E–G) when exposed to IEC-DC or directly to moDC. During direct shrimp TM exposure to moDC, the percentage of cells expressing OX40L was also increased (Figure 3D).

### 3.4. Shrimp TM Promotes a Type 2 T Cell Response via moDC

After exposing IEC-DC and moDC to chicken TM or shrimp TM, the primed moDC were collected for a subsequent coculture with allogenic naïve Th cells for 4 days. Viability was assessed after each culture step (Appendix A). The viability of IEC was minorly reduced (by 25%) due to exposure to 50 μg/mL shrimp TM, yet the viability of DC was reduced by 53%. However, shrimp TM exposure to IEC-DC did not affect the viability of the underlying moDC (Appendix A). After the moDC/T cell coculture, the T cell response was assessed based on Th subset development and cytokine secretion. The percentage of Th2 cells, based on CRTH2 expression, was not affected by coculture with chicken-TM- or shrimp-TM-primed moDC (Figure 4B). The percentage of cells expressing intracellular type 2 IL13 (Figure 4C) and IL4 secretion (Figure 4G) was enhanced in T cells cocultured with shrimp TM primed IEC-DC, while secretion of IL13 was enhanced in T cells cocultures with shrimp TM primed DC (Figure 4D). Furthermore, the proportion of CXCR3^+^ Th1 cells was decreased upon coculture with shrimp-TM-primed moDC (Figure 4E), even though the secretion of type 1 IFNγ was not significantly changed upon coculture with allergen-primed moDC (Figure 4F). Secretion of IL17 was decreased upon coculture with chicken-TM- or shrimp-TM-primed IEC-DC or moDC (Figure 4H). Furthermore, TNFα and IL21 secretion was significantly enhanced upon coculture of T cells with shrimp-TM-primed moDC (Figure 4I,J), while the secretion of regulatory IL10 (Appendix A) was not affected. Appendix A shows a heatmap of overall data.

## 4. Discussion

Shellfish allergies belong to the top eight food allergies [25] and tropomyosins are common allergenic proteins in shellfish. Although highly conserved in the animal kingdom, a clear difference between low- and high-allergenic potential is found between, respectively, vertebrate and invertebrate tropomyosins [15]. In this study, we aimed to investigate the effects of a recombinant chicken tropomyosin and recombinant shrimp tropomyosin on barrier integrity and mucosal immune activation to reveal their difference in sensitizing allergenicity. Hen’s egg-derived OVA was used as a control allergen, as it is among the top three most common food allergy inducers [2]. Previously, it was shown that OVA promotes a type 2 immune response in vitro, demonstrating its sensitizing capacity [19,26].

Proper intestinal epithelial integrity contributes to homeostasis and may prevent allergic sensitization [27,28,29]. A decrease in epithelial barrier integrity allows penetrations of allergens and potentially promotes the development of type 2 immune responses [30,31]. Next to environmental triggers disrupting the intestinal epithelial barrier and/or activating the epithelial lining, various allergenic proteins are known for their intrinsic capacity to activate epithelial cells and/or their protease activity [32,33,34]. However, recently it was established that both shrimp TM as well as OVA interact with the intestinal epithelium via upregulation of the Hippo signaling pathway, which promotes epithelial instability and production of type 2 cytokines [35]. In the current study, shrimp TM was found to reduce barrier integrity after 6 h exposure in Caco-2 cell cultures and 8 h exposure in Caco-2/HT29MTX cell cultures, which was not observed for the chicken TM or OVA. Previous studies exposing Caco-2 cells in transwell found that intestinal permeability remained intact as TEER values stayed constant, while OVA was taken up by the cells and transported over the epithelium [36,37]. Beyond barrier disruption, allergens may also activate epithelial cells to produce alarmins and inflammatory mediators that can instruct underlying immune cells. For example, Der p 10, a tropomyosin from house dust mites, is known to interact with epithelial expressed Dectin-1 [38]. Similar to Der p 10, OVA may be able to bind to Dectin-1, which normally protects against allergy development [39]. In allergic individuals, abnormal epithelial Dectin-1 expression via altered epigenetic regulation is observed [39,40], connecting epithelial functioning to allergic sensitization. Similar interactions can be hypothesized for other tropomyosins due to structural homology. However, there is no data available to verify this.

Epithelial injury often coincides with the release of alarmins such as IL33, TSLP and IL25 and pro-inflammatory IL8 secretion. In this study an increased fold change in alarmin gene expression was observed in Caco-2 cells after shrimp TM exposure (Figure 1). This is in line with the increased mRNA expression of *Tslp* and *Il33* in the intestine of mice that were sensitized against a shrimp TM (*Litopenaeus vannamei*) as described by Fu et al. [41]. Furthermore, it is known that epithelial exposure to allergenic proteins promotes the secretion of multiple cytokines, such as of IL33, IL25, TSLP and IL1α or IL1β [42,43]. Compared to Caco-2, the HT-29 cells were more responsive to the applied proteins allowing discrimination between low and high sensitizing TM (further visualized in Appendix A). Albeit from carcinogenic origin, Caco-2 and HT-29 are both commonly used model epithelial cell lines. In this study, these cell lines were used to develop the presented in vitro models, and although future studies should focus on the use of primary human intestinal epithelial cells, these cell lines possess several relevant characteristics as a model for the human intestine. Caco-2 can differentiate in culture and form ciliated polarized monolayers representing fully differentiated villus tip enterocytes. HT-29 cells do not differentiate in culture and resemble crypt epithelial cells. Therefore, these cells have different phenotypes as well as receptor expression and signaling cascades, although both resemble relevant functions of primary human intestinal epithelial cells [44,45]. This may underly the difference in sensitivity of these cell lines for activation and concomitant cytokine release upon protein exposure. Here, increased secretion of several of these cytokines from HT-29 cells was measured after exposure to shrimp TM, but not chicken TM, further emphasizing the differential epithelial interaction between the tropomyosins. The HT-29 cells were able to discriminate between proteins with high and low sensitizing allergenicity since the chicken TM, which has a low sensitizing risk, did not provoke an increase in any of the mediators. Even though it is known that allergenic proteins can induce cytokine secretion from intestinal epithelial cells, the exact interactions with the epithelium are often poorly understood [46]. Future studies could further elaborate on these effects by using primary intestinal epithelial cells or human organoids.

After IEC-DC or moDC were exposed for 48 h to chicken TM or shrimp TM, primed DC were subsequently cocultured with naïve T cells to assess the functional immune response after TM exposure. The effects of OVA exposure in this sequential mucosal immune model were already recently published, revealing type 2 immune polarization when OVA was directly exposed to DC and to a lesser extent this was also the case for IEC-DC [19]. The current study shows the differential activating capacity of shrimp TM versus chicken TM in IEC and moDC (visually summarized in Appendix A). Shrimp TM enhanced the maturation of DC, based on an enhanced frequency of DC expressing of CD80 and OX40L, and secretion of CCL20, CCL22 and IL8 by the DC. Except for the increase in OX40L expression, IEC-DC shrimp TM exposure enhanced DC activation in a similar manner. Especially, expression of OX40L by DC and secretion of CCL22 are related to promote subsequent Th2 immune development [47,48]. Although chicken TM exposure also promoted expression of CD80 and CCL20 as well as CCL22 chemokine secretion, this only occurred when moDC were directly exposed to chicken TM. In contrast to shrimp TM, chicken TM did not activate IEC-DC. This shows that also in interaction with moDC, the presence of IEC aids to differentiate between high and low sensitizing allergenicity of these tropomyosin sources.

Exposure to shrimp TM via IEC slightly altered the response of the DC compared to direct exposure of DC. However, the primed DC strongly differ in their capacity to induce T cell polarization. Shrimp TM-DC provoked increased IL13 secretion, while lowering Th1 development, whereas shrimp TM-IEC-DC induced intracellular IL13 expression and IL4 secretion in T cells. Interestingly, shrimp TM-DC enhanced proinflammatory TNFα and IL21 release on top of the type 2 immune shift after T cell coculture. A similar immune restraining effect by IEC was previously observed for OVA in this model, even though OVA was capable of activating IEC as well [19]. The current study did not assess subsequent B cell antibody production upon coculture with primed T cells. However, considering the known role of IL4 and IL13 in inducing IgE class switching [49] and based on our previous results [19], it could be hypothesized that the functional response of T cells cocultured with either shrimp TM-IEC-DC or shrimp TM-DC could be sufficient to induce IgE production in B cells. Future studies are necessary to confirm the generation of an IgE-dominant humoral response by shrimp TM.

Xu et al. exposed moDC to Bla g 7, an allergenic cockroach TM, and subsequently cocultured the moDC with T cells [50]. Bla g 7 exposure enhanced expression of Th2 driving TIM-4, CD80 and CD86 on moDC. These moDC were then capable of promoting a type 2 response during T cell coculture, which is comparable to our results for shrimp TM. Even though chicken TM was capable of activating DC (albeit less pronounced than shrimp TM), these DC were unable to instruct a type 2 response in T cells. Furthermore, beyond being capable of producing pro-inflammatory mediators, IEC are known for their capability to promote tolerance development and control intestinal homeostasis by dampening mucosal immune activation via the production of tolerogenic factors such as TGFβ or retinoic acid [51]. Indeed, as indicated in the heatmap (Appendix A), in general when compared to medium control, chicken TM did not provoke type 2 activation when exposed via epithelial cells, while shrimp TM did which may lead to allergic sensitization. This could be mediated by the secretion of tolerogenic factors, which have not been the focus of this current study. However, in the absence of epithelial cells, thus upon direct exposure of moDC to shrimp TM, a strong inflammatory type 2 response was provoked. In the current study, the dose of 50 mg/mL shrimp TM, but not chicken TM, was found to cause loss of viability in DC and IEC to some extent (Appendix A). This may have further contributed to the provocation of inflammatory responses leading to type 2 development of sequential T cell responses. In future studies, the mechanisms by which shrimp TM versus chicken TM differ in their intrinsic capacity to induce type 2 activation, including the contribution of DC maturation in presence or absence of IEC and the consequent Th2 development, should be further addressed. In addition to increased OX40L expression, Tim-4 expression may also contribute to Th2 polarization [52]. On the other hand, blocking Tim-4 on DC may lead to regulatory T cell development, and increased expression of jagged-1 may also facilitate this. The currently presented data demonstrate the differential immune activating capacity of structurally similar tropomyosins and emphasize the importance of understanding the underlying mechanism of allergic sensitization to improve allergenicity risk assessment of current and future dietary protein sources.

## 5. Conclusions

This study focused on the differential epithelial barrier disruption and mucosal immune activation by homologous tropomyosins with known differences in sensitizing allergenicity. The allergenic invertebrate shrimp tropomyosin provoked a proinflammatory response in IEC and/or DC, while disrupting epithelial barrier properties. The non-allergenic vertebrate chicken tropomyosin did not induce type 2 activation of IEC nor induced type 2 immunity. Future studies should elaborate on elucidating the mechanism underlying epithelial barrier disruption and mucosal immune activation by allergenic proteins, such as tropomyosins. This would allow for the further development of predictive tools to assess potential sensitizing capacity of food proteins.

## Figures and Tables

**Figure 1 nutrients-16-01192-f001:**
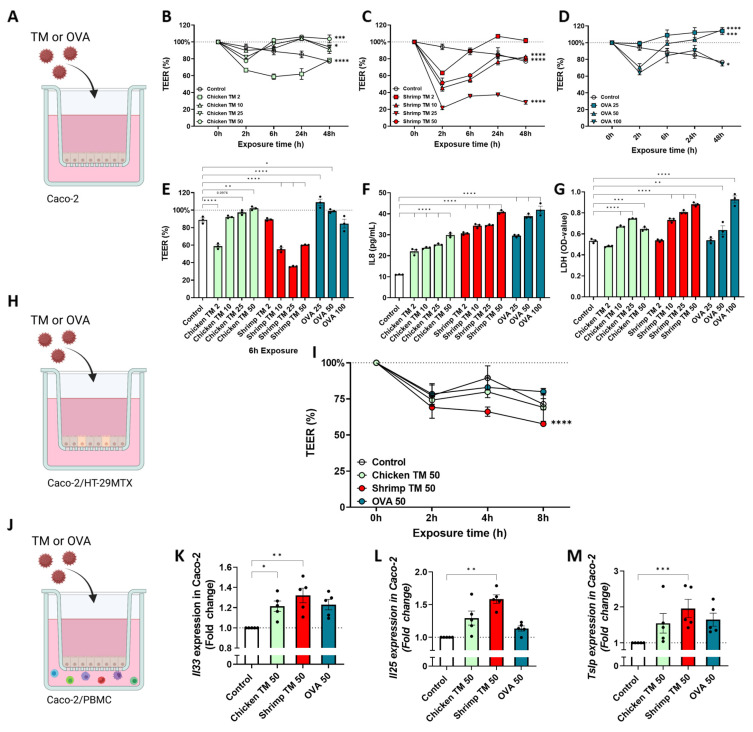
Caco-2 cells were used as a model for studying allergen-specific effects on the intestinal epithelial barrier and (**A**) they were exposed for 48 h to increasing doses of (**B**) chicken TM, (**C**) shrimp TM or (**D**) OVA while the TEER was measured at 0 h, 2 h, 6 h, 24 h, and 48 h. (**E**) TEER-values after 6 h of exposure are presented as well as (**F**) secreted IL8 and (**G**) LDH-release after 48 h of protein exposure. (**H**) Cultured Caco-2/HT-29MTX cells were exposed to 50 μg/mL chicken TM, shrimp TM or OVA for 8 h and (**I**) TEER was measured at 0 h, 2 h, 4 h, and 8 h. (**J**) A coculture of Caco-2 cells with PBMCs was performed to assess the fold change in gene expression of (**K**) Il33, (**L**) Il25, and (**M**) Tslp 8 h after 50 μg/mL chicken TM, shrimp TM or OVA exposure. Data was analyzed by Friedman test, One-Way or Two-Way ANOVA, n = 3 or n = 5, mean ± SEM (* *p* < 0.05, ** *p* < 0.01, *** *p* < 0.001, **** *p* < 0.0001).

**Figure 2 nutrients-16-01192-f002:**
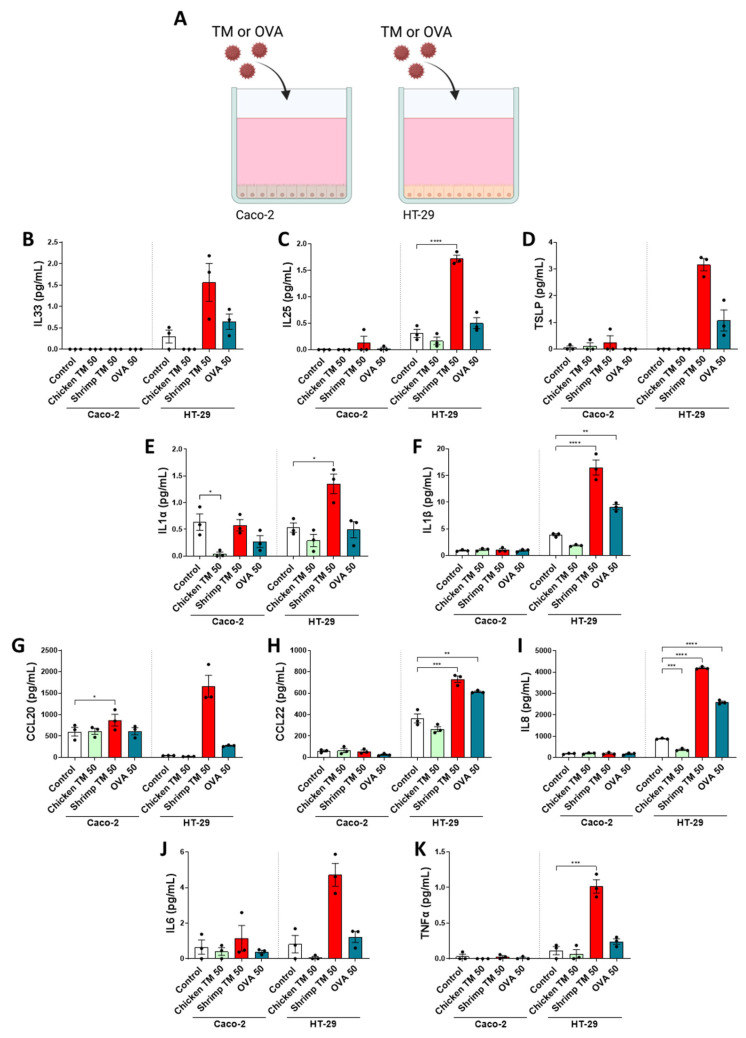
(**A**) Caco-2 and HT29 cells were cultured in 96 well flatbottom culture plates prior to 48 h exposure to 50 μg/mL chicken TM, shrimp TM or OVA. Supernatants were collected to measure (**B**) IL33, (**C**) IL25, (**D**) TSLP, (**E**) IL1α, (**F**) IL1β, (**G**) CCL20, (**H**) CCL22, (**I**) IL8, (**J**) IL6 and (**K**) TNFα secretion by multiplex array. Data was analyzed per cell line by One-Way ANOVA or Friedman test if data did not fit a normal distribution, n = 3, mean ± SEM (* *p* < 0.05, ** *p* < 0.01, *** *p* < 0.001, **** *p* < 0.0001).

**Figure 3 nutrients-16-01192-f003:**
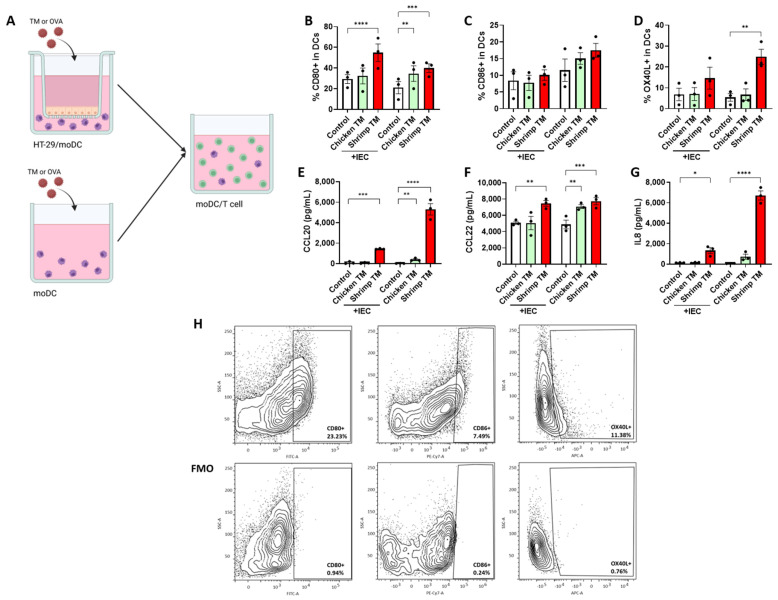
(**A**) HT-29 cells cocultured with moDC or moDC alone were exposed to 50 μg/mL chicken TM or shrimp TM for 48 h, and the primed DC were subsequently cocultured with allogenic naïve T cells for 4 days. After HT-29 cell and/or moDC coculture, expression of the costimulatory molecules (**B**) CD80, (**C**) CD86 and (**D**) OX40L was measured by flow cytometry. In addition, supernatant concentrations of (**E**) CCL20, (**F**) CCL22, and (**G**) IL8 were measured. (**H**) The flow cytometry gating strategy is given with a representative sample and corresponding FMOs. Data is analyzed by One-Way ANOVA, n = 3, mean ± SEM (* *p* < 0.05, ** *p* < 0.01, *** *p* < 0.001, **** *p* < 0.0001).

**Figure 4 nutrients-16-01192-f004:**
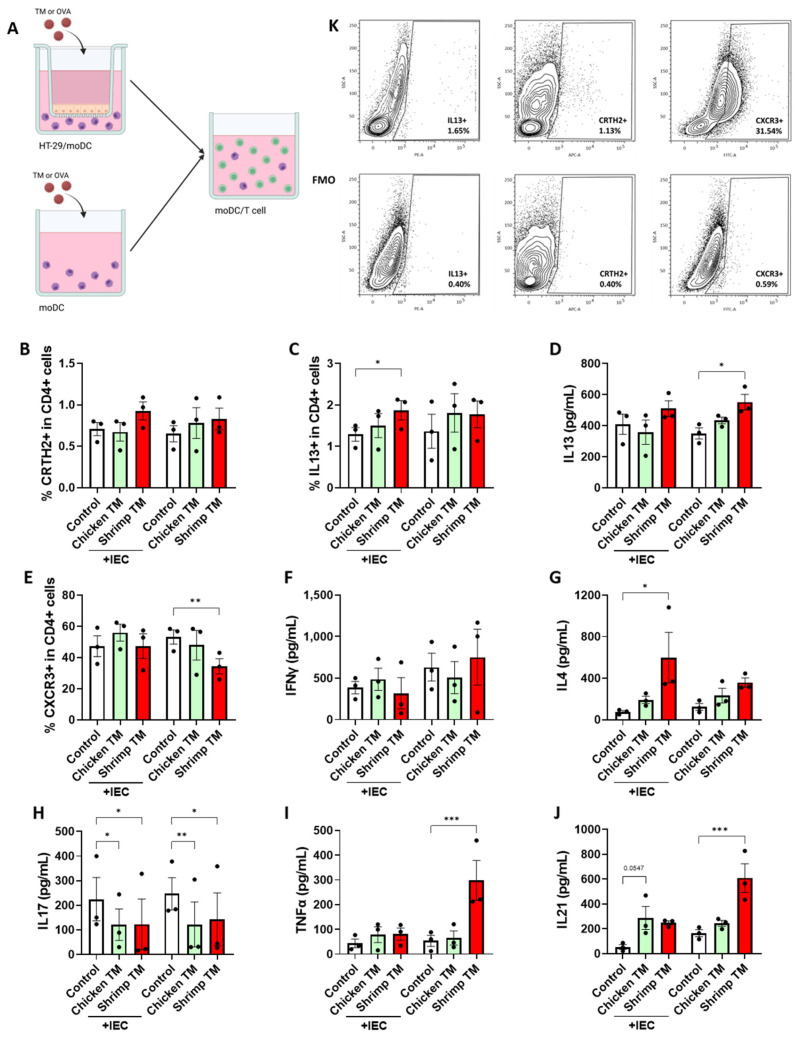
(**A**) HT-29 cells cocultured with moDC or moDC alone were exposed to 50 μg/mL chicken TM or shrimp TM for 48 h, and the primed DC were subsequently cocultured with allogenic naïve T cells for 4 days. Cocultured DC and klT cells were collected to analyze T cell subsets and supernatants were collected to measure secreted cytokines. The percentage of (**B**) CRTH2 and (**C**) IL13 expressing T cells as well as the level of secreted (**D**) IL13 and (**G**) IL4 were measured as part of the type 2 response. The percentage of (**E**) CXCR3 expressing T cells and (**F**) IFNγ was measured as part of the type 1 response. Furthermore, secreted (**H**) IL17, (**I**) TNFα, and (**J**) IL21 were measured. (**K**) The flow cytometry gating strategy is given with a representative sample and corresponding FMOs. Data is analyzed by One-Way ANOVA, n = 3, mean ± SEM (* *p* < 0.05, ** *p*< 0.01, *** *p* < 0.001).

**Table 1 nutrients-16-01192-t001:** Primer pair sequences for gene transcription analysis in Caco-2 cells. *fw*, forward; *rv*, reverse.

Gene	Primer Pairs	Reference	Cycling Conditions
*Gapdh*	*fw* 5′ GAAGGTGAAGGTCGGAGTCAA 3′ *rv* 5′ ACGTACTCAGCGCCAGCATC 3′	[22]	Pre-incubation 2 min 50 °C Incubation 10 min 95 °C 40 cycles
*Il33*	*fw* 5′ GAGCTAAGGCCACTGAGGAA 3′ *rv* 5′ TGGGCCTTTGAAGTTCCATA 3′	[23]	
*Il25*	*fw* 5′ CCAGGTGGTTGCATTCTTGG 3′ *rv* 5′ TGGCTGTAGGTGTGGGTTCC 3′	[24]	
*Tslp*	*fw* 5′ CTCTGGAGCATCAGGGAGAC 3′ *rv* 5′ CAATTCCACCCCAGTTTCAC 3′	[22]	

## Data Availability

The data presented in this study are available on request from the corresponding author.

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
