# Peer review of "Allergenic Shrimp Tropomyosin Distinguishes from a Non-Allergenic Chicken Homolog by Pronounced Intestinal Barrier Disruption and Downstream Th2 Responses in Epithelial and Dendritic Cell (Co)Culture"

_nutrients, 2024, doi:10.3390/nu16081192_

Round 1

Reviewer 1 Report

Comments and Suggestions for Authors

A well designed study to investigate possible explanations for differently allergenic proteins. I agree with the authors discussion, but like to emphasize one point: the study cultures moDC and T cells directly with TM which would require the intact protein to cross the epithelial barrier. Although I consider this possible in inflamed mucosa there still remains the problem of hen and egg: is inflammation necessary for TM to ross the barrier, or is the epithelial barrier sufficiently permeable to TM to allow its uptake in sufficient amount to trigger a small inflammatory response which then will induce its own growth? In practice I do not consider this a major poiont since I assume some local inflammation to be present, for pathophysiological explanations it bears a larger weight. The authors could have strengthened this point by measuring the possible TM uptake in their barrier model (maybe it already has been published then it should be referenced.

A number of language mistake (capitalizing incorrectly) betrays the writer of the original draft – they should be eliminated.

Comments on the Quality of English Language

A few capitalizing mistakes - and I have been taught that "however" always is followed by a comma.

Author Response

Dear Reviewer 1,

We would like to thank the Reviewer for the feedback that has been provided.

With regards to the question raised on epithelial permeability and crossing of the TMs, the data in this manuscript demonstrates that only shrimp TM but not chicken TM or OVA can disturb the epithelial barrier integrity. However, bot shrimp TM and OVA are able to induce an inflammatory response from the epithelium and during subsequent cocultures. However, previous studies have demonstrated that a decrease in epithelial barrier integrity (measured by TEER) is not a prerequisite for of allergens transport across the epithelium and/or enhanced expression of alarmins as was previously demonstrated for Pru p 3 by Tordesillas et al. 2023 (DOI: 10.1111/cea.12202) and for ovalbumin by Khuda et al. 2021 (DOI: 10.1002/mnfr.202100576). However, it has also been described that other tropomyosins, such as Pen j 1, cross the epithelial barrier while increasing the epithelial permeability (Kunimoto et al. 2011 (DOI: 10.1271/bbb.110021)). So, although disturbing epithelial barrier integrity may contribute to the transport of allergens across the epithelium, this disturbance is not required for allergens to become available in the underlying mucosa. It could be hypothesized that the inflammatory response from the epithelium plays a more determining role in subsequent mucosal immune activation, however based on only the currently presented data this cannot be concluded and future study are necessary to further confirm this.

Furthermore, we have aimed to correct for the capitalizing mistakes that were made, several changes were made and are visible via the ‘Track Changes’ option.

Reviewer 2 Report

Comments and Suggestions for Authors

Marit Zuurveld et al. submitted an original article entitled allergenic shrimp tropomyosin distinguishes from a non-allergenic chicken homolog by pronounced intestinal barrier disruption and downstream Th2 responses in epithelial and dendritic cell (co)culture.
Nowadays, food allergic diseases are a growing health problem. Tropomyosins derived from vertebrates are non-allergenic, but invertebrate homologs can be pan-allergens.  The authors aimed to compare the risk of sensitization between chicken tropomyosins and shrimp tropomyosins through affecting intestinal epithelial barrier integrity and type mucosal immune activation.
In this order, they studied epithelial activation and barrier effects upon exposure to 2-50 μg/mL chicken tropomyosins, shrimp tropomyosins, and ovalbumin as control allergen via Caco-2, HT-29MTX, or HT-29 intestinal epithelial cells. They determined intestinal barrier integrity, gene expression, cytokine secretion, and immune cell phenotypes in suitable human in vitro models. Finally, the authors concluded that shrimp (not chicken) tropomyosins „disrupted the epithelial barrier while triggering type 2 mucosal immune activation, both key events in allergic sensitization”.
This project is interesting and well-fitting into the niche. In vitro, models enable to discriminate between proteins in food products with low or high risk for allergic sensitization.
The manuscript is well prepared and could be interesting for the reader of this Journal. I have only one suggestion: Fig. 2 and 4 should be corrected. They contain too much information and labels are unreadable.

Author Response

Dear Reviewer 2,

We would like to thank the Reviewer for the feedback that has been provided.

Changes have been made to the layouts of Figure 2 and 4 to improve the readability of the labels.